# Higher-order quantum spin Hall effect in a photonic crystal

Biye Xie [1,2,6], Guangxu Su [1,3,6], Hong-Fei Wang [1,2,6], Feng Liu [4], Lumang Hu [1,3], Si-Yuan Yu [1,2], Peng Zhan [1,3✉], Ming-Hui Lu [1,2,5✉], Zhenlin Wang [1,3✉] & Yan-Feng Chen [1,2✉]

The quantum spin Hall effect lays the foundation for the topologically protected manipulation of waves, but is restricted to one-dimensional-lower boundaries of systems and hence limits the diversity and integration of topological photonic devices. Recently, the conventional bulk-boundary correspondence of band topology has been extended to higher-order cases that enable explorations of topological states with codimensions larger than one such as hinge and corner states. Here, we demonstrate a higher-order quantum spin Hall effect in a two-dimensional photonic crystal. Owing to the non-trivial higher-order topology and the pseudospin-pseudospin coupling, we observe a directional localization of photons at corners with opposite pseudospin polarizations through pseudospin-momentum-locked edge waves, resembling the quantum spin Hall effect in a higher-order manner. Our work inspires an unprecedented route to transport and trap spinful waves, supporting potential applications in topological photonic devices such as spinful topological lasers and chiral quantum emitters.

[1] National Laboratory of Solid State Microstructures, Collaborative Innovation Center of Advanced Microstructures, Nanjing University, Nanjing 210093, China. [2] Department of Materials Science and Engineering, Nanjing University, Nanjing 210093, China. [3] School of Physics, Nanjing University, Nanjing 210093, China. [4] Department of Nanotechnology for Sustainable Energy, School of Science and Technology, Kwansei Gakuin University, Sanda 6691337, Japan. [5] Jiangsu Key Laboratory of Artificial Functional Materials, Nanjing 210093, China. [6] These authors contributed equally: Biye Xie, Guangxu Su, Hong-Fei Wang. ✉email: zhanpeng@nju.edu.cn; luminghui@nju.edu.cn; zlwang@nju.edu.cn; yfchen@nju.edu.cn

Trapping and guiding the flow of light lie at the heart of modern integrated photonic devices that are crucial for the realization of the photonic quantum communication and computing[1–3]. However, owing to unavoidable disorder and imperfections during the fabrication process, the localization and propagation of light in traditional photonic devices suffer from fragility and backscattering. Fortunately, by invoking the spin (or pseudospin) degree of freedom of photons[4–8] combined with a nontrivial band topology, we can enable spinful light to propagate unidirectionally without backscattering, which is now known as the quantum spin Hall effect (QSHE) of light[9–11]. This effect which describes boundary states of a sample is characterized by a nontrivial topological invariant and support boundary spin (or pseudospin) transport, yielding the realization of photonic topological insulators and underpining the development of many spinful photonic devices[12–20]. However, the conventional QSHE of light only focuses on the one-dimensional propagation which may put restrictions on the diversity and integration of photonic devices. Explorations of the multi-dimensional manipulation of topological protected waves, especially in a single structural design will significantly promote the versatility and may facilitate the development of integrated photonics.

Recently, higher-order topological insulators (HOTIs) have been introduced as new kinds of topological phases of matter which go beyond the conventional bulk-boundary correspondence[21–33]. If we define the codimension $l$ of an $n$-dimensional ($n$D) state in an $m$D system as $l = m − n$, then an $l$th-order topological insulator can be defined as a topological insulator with $l$-codimensional boundary states. These lower-dimensional boundary states such as hinge states and corner states provide new degrees of freedom with which to manipulate waves and support integrated topological devices. Previous experimental realizations of HOTIs were all limited to explorations of the existence and robustness of these boundary states[27–37]. The internal degrees of freedom of waves, such as the spin and pseudospin, offer a new dimension with which to investigate the wave physics and sustain vast applications from the information processing to the topological quantum computing which have so far not been intertwined with the higher-order topology.

In this article, we propose a spinful photonic second-order topological insulator. Moreover, we demonstrate that corner states are induced and separated from bulk and edge states by the filling anomaly[38] in the $C_{6v}$-symmetric topological crystalline insulator as shown in Fig. 1a. Intriguingly, we show that corner states in our implementation have pseudospins (denoted as spinful corner states) which are realized by the linear combination of $d$ states in the folded band structure as shown in Fig. 1b. As a consequence, when the pseudospin–pseudospin coupling is considered in a finite-size structure, the pseudospins will have nonzero polarizations[39]. To observe these nonzero pseudospin polarizations, we experimentally fabricate a sample comprised of a second-order photonic topological insulator and a trivial topological insulator as shown in Fig. 1c. We then apply a excitation source with pseudospin at the middle of the interface between two topologically distinct areas and observe the near-field electromagnetic wave. We find that the wave is guided to and localized at different corners with opposite pseudospin polarizations, demonstrating a higher-order QSHE as shown in Fig. 1c.

## Results

### Lattice structure and nontrivial higher-order topology. The photonic crystal (PC)[40,41] we considered here is a triangular lattice of hexagonal clusters with six dielectric rods in each cluster as shown in insets of Fig. 2a. For simplicity, we only consider

transverse magnetic (TM) modes. The distances among the dielectric rods determine the coupling strength and hence there are two competing parameters: the inter-cell couplings $t_{inter}$ and the intra-cell couplings $t_{intra}$ that determine the band structure of the PC. If $t_{inter} = t_{intra}$, the PC has a honeycomb lattice structure and the band structure of the triangular lattice is the band-folding version of the one in honeycomb lattice (The Dirac cones at Brillouin zone corners $K$ and $K'$ of honeycomb lattice are folded to the Brillouin zone center $\Gamma$ of the triangular lattice)[9,39]. By adjusting inter-cell couplings and intra-cell couplings, the Dirac cone at $\Gamma$ can be gapped out and a topological phase transition occurs when the PC evolves from $t_{inter} < t_{intra}$ (denoted as the expanded lattice which is topologically nontrivial) to $t_{inter} > t_{intra}$ (denoted as the shrunken lattice which is topologically trivial) (see Supplementary Note 1). In our study, we set $a = 18.0$ mm, $r = 1.8$ mm, $h = 25.0$ mm, $\epsilon = 8.5$ as lattice constant, radius of rods, height of the rods and relative dielectric constant of the rods, respectively, for all lattices. $b_1 = 5.0$ mm and $b_2 = 6.8$ mm are distances between nearest-neighbor rods in the unit cell which are proportional to the inter-cell coupling strengths for the ordinary insulator (OI) and topological insulator (TI), respectively. (The intra-cell coupling strength is automatically determined once the inter-cell coupling strength is determined since the lattice is the tessellation of the unit cell.)

For the expanded lattice case, the PC has a dimensional hierarchy of the higher-order topology in which both the first- and the second-order photonic topological insulator emerge[32,38]. The first-order and the second-order topologically insulating phases are characterized by the bulk polarization $\mathbf{P} = (P_x, P_y)$ and the secondary topological index $Q^c$, respectively. For the topologically nontrivial configuration which belongs to $h_{3c}^{(6)}$ configuration, we have $\mathbf{P} = (0, 0)$ and $Q^c = \frac{1}{2}$ which indicate a vanishing dipole moment and a nontrivial second-order topologically insulating phase (see details in the Methods section and Supplementary Note 2). Besides the $h_{3c}^{(6)}$ case, there is another coupling configuration denoted as the $h_{4b}^{(6)}$ case for the $C_6$ symmetric higher-order topological crystalline insulator. However, due to the nonnegligible next-nearest-neighbor coupling in the photonic crystals of $h_{4b}^{(6)}$ case, there is no full bandgap and thus we do not take it into consideration here (see Supplementary Note 3).

**Pseudospin and the bulk–edge–corner correspondence.** Besides the existence of the higher-order topology, this staggered couplings between inter-cell and intra-cell sites in the enlarged unit cells which have more internal degree of freedom, introduce an extra pseudospin degree of freedom as first proposed by Wu and Hu in ref. [9]. Specifically, the enlargement of unit cell introduces additional interal degrees of freedom that can be regarded as pseudospins under the constraint of $C_{6v}$ point group symmetry and time-reversal symmetry. For example, at the Brillouin zone center, the unit cell has $p_x(p_y)$ and $d_{xy}(d_{x^2−y^2})$ orbitals (see Supplementary Note 1). The linear combinations of these two sets of orbitals

$$p_\pm = (p_x \pm ip_y)/\sqrt{2}, d_\pm = (d_{xy} \pm id_{x^2−y^2})/\sqrt{2} \qquad (1)$$

form pseudospins with pseudo-time-reversal symmetry defined as $T = U\kappa$ as shown in Fig. 1b, c. Here $U = [D_{E'}(C_6) + D_{E'}(C_6^2)]/\sqrt{3} = −i\sigma_y$ and $D_{E'} = [1/2, −\sqrt{3}/2; \sqrt{3}/2, 1/2]^T$ represents the irreducible representation in the $C_6$ symmetry group and $\kappa$ is the complex conjugation operator. We note that $T^2 = −1$ and thus the Kramers degeneracy is realized. If we juxtapose the topologically nontrivial lattice (TI) with the

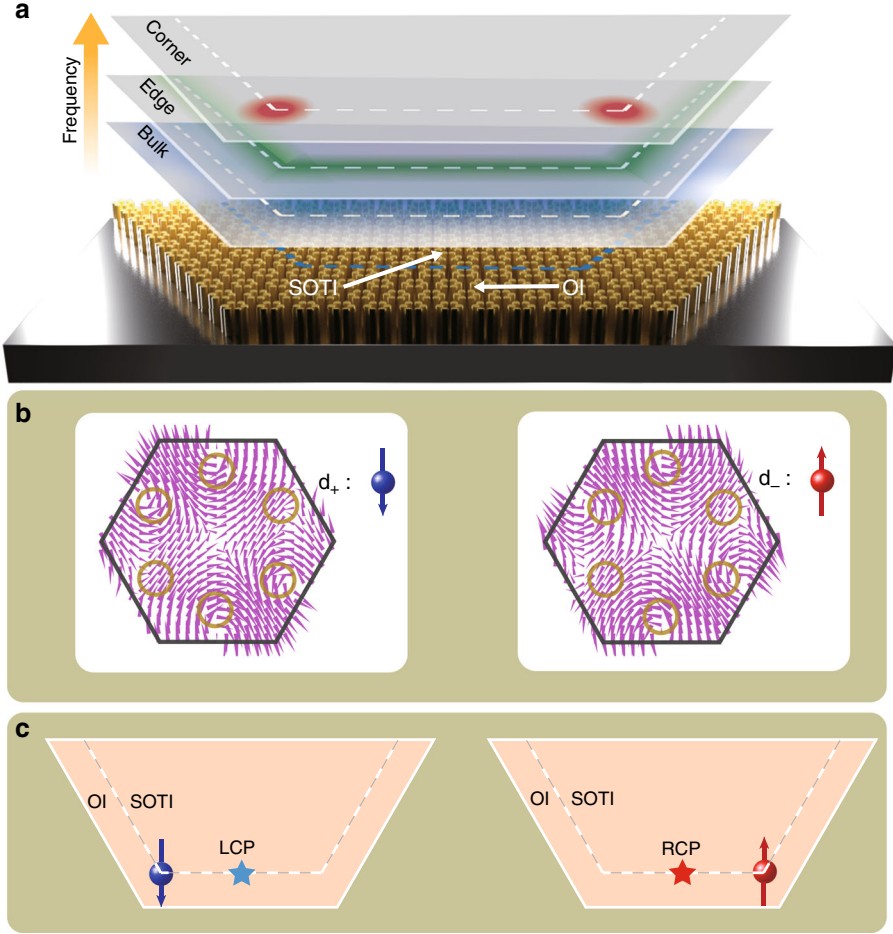

**Fig. 1 Higher-order quantum spin Hall effect in a photonic crystal. a** The dimensional hierarchy of the higher-order topological insulator in a dielectric photonic crystal. Corner, edge and bulk states are represented by blue, green and red colors, respectively, and separated from each other in the frequency domain. **b** Two pseudospins defined by the in-plane magnetic field (represented by the purple arrows) configuration in the unit cell. **c** The scheme of achieving directional localization of psedospin-polarized corner states excited by a pseudospin-dependent source. The blue (red) star represents the position of a left circular (right circular) polarized light as a pseudospin-dependent source. The blue (red) sphere represents the position of the corner states with a pseudospin up (down) polarization (represented by arrows).

topologically trivial lattice (OI) (see Fig. 2a and Supplementary Note 4), there exists 1D spinful edge states which mimic the 2D quantum spin Hall effect as shown in Fig. 2b–d. Due to the spin-momentum locking, the spinful edge states can be selectively excited and propagate unidirectionally along the 1D boundary as shown in Fig. 2c.

Next we study the second-order topological insulator (denoted as SOTI which shares the same parameters with TI in Fig. 2a) with corner states by considering a 2D photonic lattice as shown in the lower inset of Fig. 2e in which the SOTI lattice is surrounded by the OI lattice. The eigenvalues of TM modes are numerically calculated as shown in Fig. 2e. We observe a dimensional hierarchy of higher-order topology as both 1D edge states (represented by yellow dots) and 0D corner states (represented by blue dots) emerge between 2D bulk states (represented by gray dots). As shown in the right panel of Fig. 2e, the local field intensity is much lower in the bandgap than those in the bulk, edge, and corner states. The measured field intensity fits well with the numerical simulation as shown in the right panel. We also observed the discrete distribution of edge and bulk states which is induced by the finite-size effect. Due to the closed boundary, the spectrum measured at the right corner is almost the same as the one measured at the left corner (see Supplementary Note 8). The corner states here can be regarded

as boundary states of the gapped 1D edge Hamiltonian. We further measured the transmission spectra as shown in the right panel of Fig. 2e which matches well with the numerical result. The field distributions of corner states are numerically calculated and shown in Fig. 2f (lower panel). We find that the out-of-plane electric field is strongly localized at six corners and exponentially decay away from the corners, demonstrating its 0D geometry. We fabricate this hexagonal photonic lattice by using the alumina rods arranged in the same way and excite the corner state with a source at frequency $f_c = 9.760$ GHz. The field distribution is measured by using the near-field scanning technique which matches well with the simulation as shown in Fig. 2f (upper panel) (see other field distributions of corner states in see Supplementary Note 5).

**Pseudospin-polarized corner states**. Due to extra degrees of freedom of the enlarged unit cell, the topological corner states also possess the pseudospin of Fig. 1b. Because of pseudospin–pseudospin couplings, pseudospin polarizations of corner states are not zero. Therefore they can be selectively excited by a source field with pseudospin. We demonstrate the pseudospin structures of corner states as shown in Fig. 3. We use the half-hexagonal structure to avoid whisper-gallery modes in a

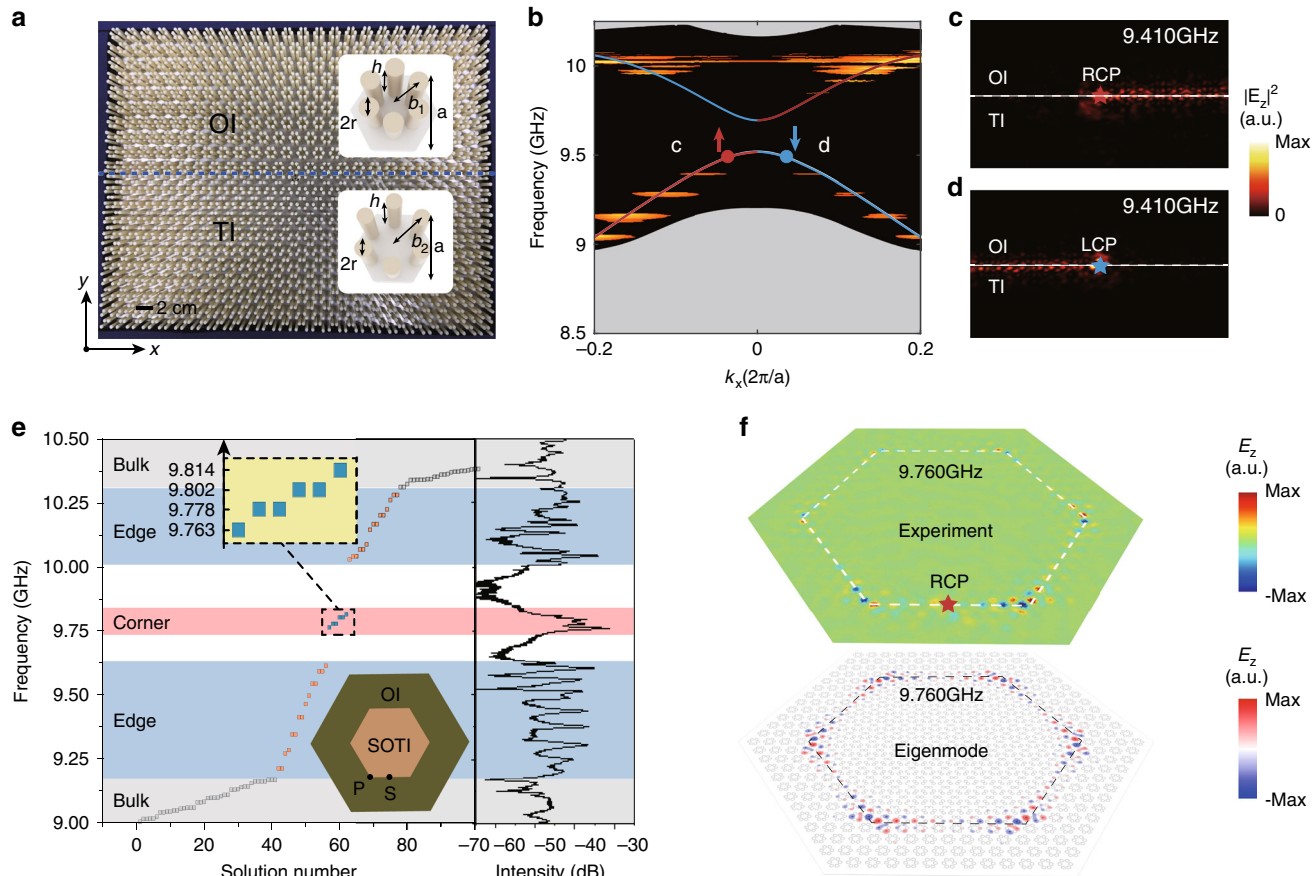

**Fig. 2 Emergence of multi-dimensional boundary states at the interface between two topologically distinct photonic crystals. a** The combination of an ordinary insulator and a topological insulator with a line-shape boundary (blue dashed line). **b** The measured (yellow area) and the simulated (solid line) spinful edge states in the projected band structure. The unidirectional propagation of light to the **c** right and **d** left directions with an right circular polarized (RCP) and a left circular polarized (LCP) source, respectively. **e** The numerically calculated eigenvalues (left panel) of and measured transmission (right panel) of the hexagonal meta-structure (lower inset). The bulk, edge and corner states are represented by gray, blue and yellow dots, respectively. **f** The field distribution of the numerically calculated (lower panel) and experimentally excited (upper) corner states at 9.760 GHz.

closed boundary as shown in Fig. 3a (upper panel) (see the tight-binding model which mimics this photonic crystal in Supplementary Note 6). The SOTI and OI share the same parameters as those in Fig. 2a. The pseudospin-dependent source is realized by using three point sources with each one differing in phase by $2\pi/3$, as shown in Fig. 3a (lower panel). This source has a nonzero orbital angular momentum (OAM) and is used as the pseudospin-dependent source. The local field intensity is acquired by taking the averaged value of experimental data of nine adjacent spatial points centered on the dielectric rods (see Supplementary Note 7). The numerically calculated eigenmodes are presented in Fig. 3b. We note that there are two near-degenerate mid-gap states which are localized corner states. For those corner states we display their pseudospin textures in Fig. 3c, d. For the corner state with lower frequency at 9.775 GHz, the pseudospin polarization of the left and right corners are $d_+$ and $d_-$, respectively (denoted as the eigenmode $\phi_1$). While for the corner state with higher frequency of 9.800 GHz, the pseudospin polarization of the left and right corners are both $d_+$ (denoted as the eigenmode $\phi_2$). The phases of these two corner states are constant (with a change of signs) and have no vortex structure (see Supplementary Note 9). We note that there is not a case of corner state with both $d_-$ at two corners. This is because the pseudo-time-reversal symmetry which is related to the $C_6$ symmetry is broken at the corners of the sample. We also calculated the frequencies, the field distributions of all the corner states from the analytic tight-binding

model which mimics our photonic crystals (see Supplementary Note 6). Owing to the existing sublattice symmetry in the tight-binding model, the corner states are mixed with the edge states. Nevertheless, field distributions of all corner states match well between the analytic model and the photonic crystal. To experimentally observe pseudospin polarization of corner states, we put the OAM source in the middle of the 1D boundary between two corners. The simulated and measured local field intensities are obtained and depicted in Fig. 4a–d by putting a probe at the left (denoted as A) corner and the right (denoted as B) as shown in Fig. 3a. We observe an obvious difference of the local field intensity between two opposite pseudospin excitations at two corners as shown in Fig. 4. For left circular polarized (LCP) excitation we find the A corner has a relatively higher local field intensity than the B corner while for right circular polarized (RCP) excitation, the field is mainly localized at the B corner. We note that there is only one peak for these two eigenmodes. This is induced by linear combinations of previous two eigenmodes $\phi_1$ and $\phi_2$ as $\phi_{LCP} = \phi_1 + \phi_2$ and $\phi_{RCP} = \phi_1 - \phi_2$.

For further confirmation of the localization of corner states, we map out the out-of-plane electric field intensity $|E_z|^2$ by applying the near-field scanning method as shown in Fig. 4e, f. We find that the field is exponentially localized at the left corner for LCP source while at the right corner for the RCP source for both simulations (white panels) and experiments (black panels). This directional localization of waves at different corners with respect

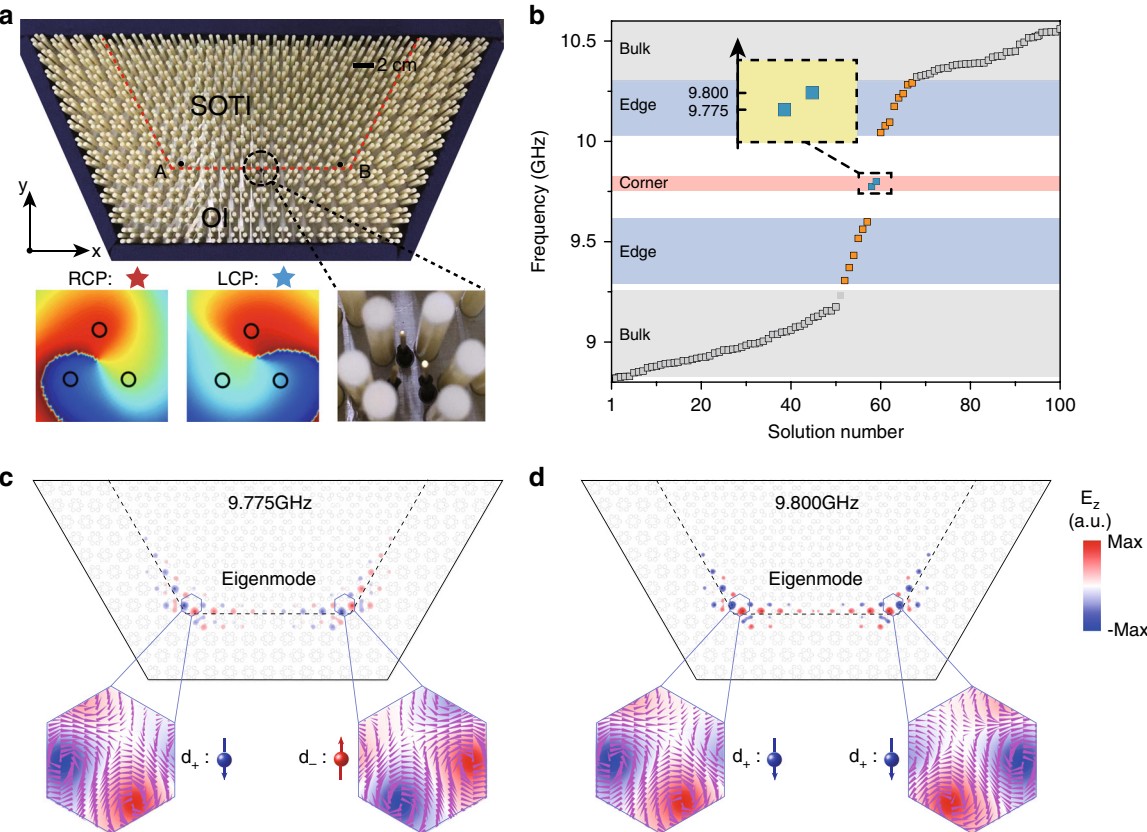

**Fig. 3 The pseudospin-polarized corner states. a** The half-hexagonal meta-structure consists of an second-order topological insulator (SOTI) and an ordinary insulator (OI) (upper panel). The pseudospin-dependent source are realized by three point sources with each phase differing by $2\pi/3$. The experimental measured field distribution of the source exhibit orbital angular momentum fields (lower panel). **b** The calculated eigenmodes of the half-hexagonal meta-structure. **c** The pseudospin polarizations represented by configurations of the in-plane magnetic field (purple arrows) at two corners are opposite and the same for **c** the lower frequency mode and **d** the higher frequency mode, respectively.

to opposite pseudospin-dependent sources demonstrates a photonic higher-order quantum spin Hall effect. As shown in Fig. 4e, f, we demonstrate a directional localization of topological polarized microwaves. This may enable robust and controllable microwave polarized probe and imaging. When considering the out-of-plane radiation, we may realize a topological microwave antenna that has the directional selection of emission of signals with a high-quality factor, low-Ohmic loss, and topological robustness. We note that there is small shift in the frequencies between the simulated and measured data which is due to the precision of the fabrication.

## Discussion

In conclusion, we theoretically propose and experimentally realize a spinful higher-order photonic topological insulator with both 1D edge states and 0D corner states are observed. Moreover, We achieve the directional localization of pseudospin-polarized corner states for microwave radiation, which are the photonic higher-order version of the QSHE. Although the existence of the corner states is guaranteed by the nontrivial higher-order topology, the frequencies and linear combinations of the corner states are influenced by the finite-size effect and the precision of the fabricated samples. The difference in the frequencies between the simulation and experiment are due to the finite height of the dielectric rods in the experiment. Our work opens the way to study spinful photonic HOTIs and supports explorations in lower-dimensional spintronics and spin photonics. The directional localization of the microwave radiation in 0D enables the designs of topological optical switches, energy splitters,

topological lasings, and unidirectional light trappings[42–45]. Moreover, since our implementation is simple and based on the dielectric material, this work can be directly extended to optical frequencies using silicon-based coupled resonator optical waveguides[14] and femto-lasing direct writing waveguides[28]. It would then be possible to explore the quantum behavior of single or multiple photons which are related to the pseudospin degree of freedom, such as the spinful quantum walk, spinful quantum correlation, and entanglement.[46–49] When coupled to quantum emitters, the spinful corner states may support chiral emission of photons with potential applications in quantum simulations[47]. Although our implementation is based on photonic crystals, we expect to observe a similar effect in other periodic classical wave systems such as the phononic crystals[50–53], and with excitations such as, surface plasmon polariton, and in electronic circuits[24,54].

## Methods

**Design of the second-order spinful photonic topological insulator**. We design our 2D photonic crystals by using dielectric cylinders consisting of alumina ceramics. Cylinders are stuck to the metallic plates with a depth of 1.5 mm. Six dielectric rods in a unit cell support multipolar resonant modes at different frequencies such as the monopole, dipole, quadrupole and so on, mimicking atomic orbitals. We use the absorbing material to surround our sample to reduce wave reflections and background noise. We note that there are two differences between our design and the corresponding tight-binding model. First, in the low frequency limit, our photonic crystals can be regarded as homogeneous materials and their dispersion relations are linear which naturally break the chiral symmetry (sublattice symmetry) and lead to a frequency shift of all states (bulk, edge, corner states) comparing to the tight-binding models. Second, the pseudospin in our implementation is defined as a continuous configuration (state) of in-plane magnetic field while this is not the case for the tight-binding model. For the tight-binding model (TBM), the vacuum is a topologically trivial insulator for electrons

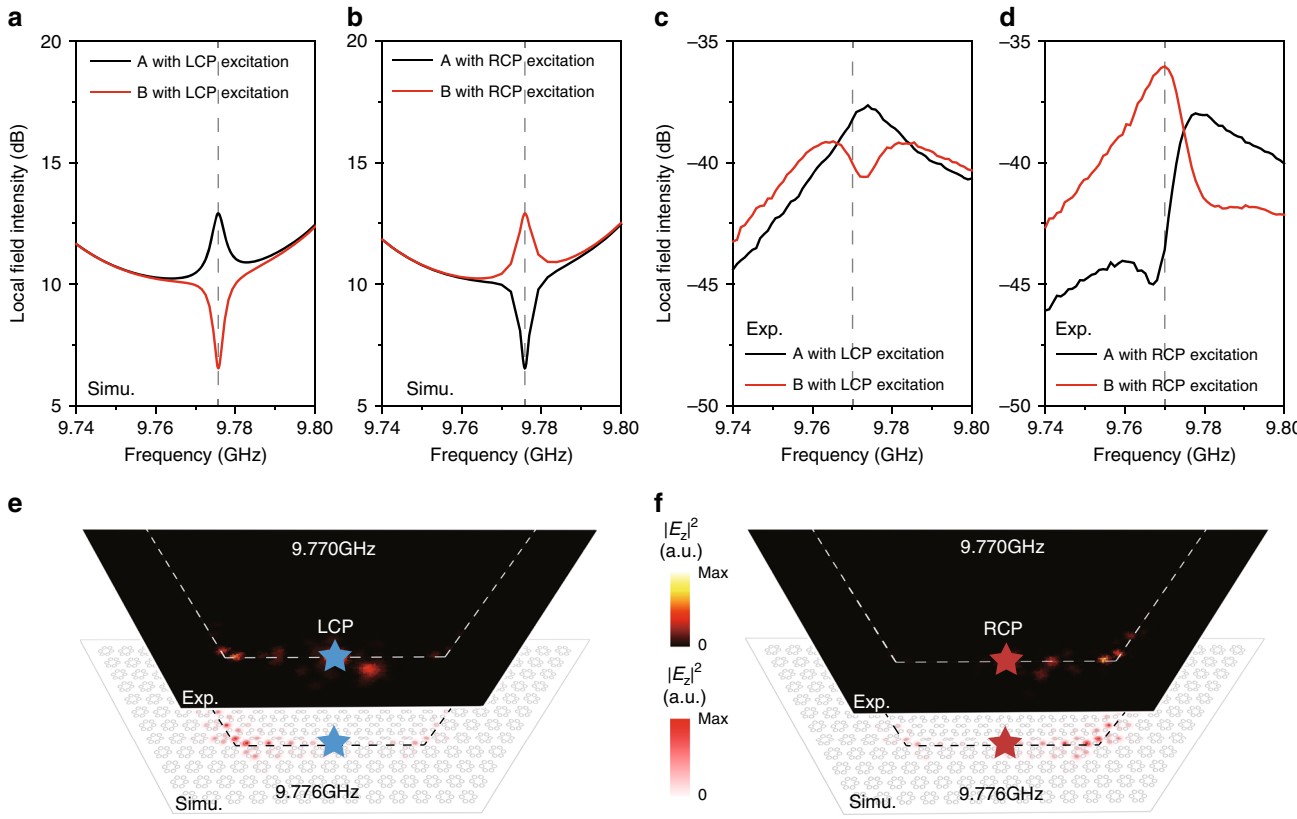

**Fig. 4 Directional localization of light at corners.** The **a** (**b**) simulated and **c** (**d**) experimentally measured transmission spectrum excited by an LCP (RCP) source located at the center of the 1D, respectively. **e** The simulated (white panel with excitation source at 9.776 GHz) and experimentally measured (black panel with excitation source at 9.770 GHz) $|E_z|^2$ distribution for a LCP excitation. **f** The simulated (white panel with excitation source at 9.776 GHz) and experimentally measured (black panel with a excitation source at 9.770 GHz) $|E_z|^2$ distribution for a RCP excitation.

and therefore it naturally characterizes a closed system unless we introduce non-Hermitian loss at boundaries of the TBM. However, for photonic crystals, the vacuum (or the air) is a conductor for electromagnetic waves and we should use an insulating photonic crystal to encircle the topological one. In our implementations as shown in Fig. 2e (closed boundary) and Fig. 3b (open boundary) in the main text, although the latter one has an open 1D boundary, it will not affect the existence of the 0D corner states as the corners are well surrounded by insulating photonic crystals. This has been proved by comparing Fig. 2e in the main text and Supplementary Fig. 8 (see Supplementary Note 7) where they are similar to each other.

**Higher-order topological invariants.** Based on the eigenvalues of the $C_6$ operation at high symmetric points in the first Brillouin zone, the bulk polarization and the secondary topological indice are obtained as follows[38],

$$P_x = P_y = [K_1] + 2[K_2] = 0 \quad \text{mod} \quad 1, \tag{2}$$

$$Q^c = \frac{1}{4}[M_1] + \frac{1}{6}[K_1] \quad \text{mod} \quad 1 \tag{3}$$

where $[\Pi_p] = \#\Pi_p - \#\Gamma_p$ and $\#\Pi_p$ is defined as the number of bands below the bandgap with rotation eigenvalues $\Pi_p = e^{\frac{2\pi i(p-1)}{6}}$ for $p = 1, 2, 3, 4, 5, 6$. $\Pi$ stand for high symmetric point K, M, and $\Gamma$ (see Supplementary Note 1). The primitive generator of the topologically nontrivial configuration of our PC is $h_{3c}^{(6)}$ (see Supplementary Note 2).

Based on the theory of topological crystalline insulators, we have $[M_1] = 2$ and $[K_1] = 0$. Therefore

$$p_i = 0, Q^c = \frac{1}{2} \tag{4}$$

We note dipole moments are always vanishing for $C_{6v}$-symmetric lattice and the corner topological index is $Q^c = \frac{1}{2}$, indicating a nontrivial second-order topological insulating phase in our PC.

**Numerical simulations.** We use commercial software: COMSOL MULTIPHYSICS to conduct all numerical simulations of our samples. In all simulations, we use 2D photonic crystals to mimic our sample. This is valid when we consider TM modes.

For boundary conditions of the ribbon structure we used to obtain Fig. 2b, we set the in-plane boundary parallel to the propagation direction as the perfect electric conductor (PEC) and the boundary perpendicular to the propagation direction as Floquet periodic boundary (see Supplementary Note 3). In simulating Figs. 2e, f, 3b, and 4e, f, we use the scattering boundary condition for the boundaries parallel to the interface of two PCs.

In simulating Fig. 3b, we find extra mid-gap states which are upper boundary states and irrelevant to our study. Therefore, we use an algorithm to eliminate those irrelevant states as follows: We take the upper boundary of the nontrivial region as an area of $18a \times 2.5l$, where $a$ is the lattice constant and $l = a/\sqrt{3}$ is the edge length of the unit cell. By defining the proportion of the boundary field in the whole region, the mode independent of the boundary can be selected. The filter condition is defined as $\int_{\text{edge}}|E_z|^2 ds / \int_{\text{all}}|E_z|^2 ds < 0.5$. This ensures that in the eigenfrequency of Fig. 3b in the main text, the edge mode at the upper boundary is naturally eliminated. For the calculation of pseudospins in Fig. 3b, we first obtain the out-of-plane electric field and then directly calculate the corresponding in-plane magnetic field.

**The pseudospin-dependent source.** In our implementation, we use an OAM source as the pseudospin-dependent source. To show the overlap between the source excitation and pseudospin polarization, we numerically calculate the overlap integrals defined as $|\langle d_\pm | \phi \rangle|$. Here $d_\pm$ is the pseudospin eigenstates and $\phi$ is the OAM field in a unit cell. For a source with clockwise/anti-clockwise circular polarization at 9.755 GHz, we extract the distribution of excitation field over a unit cell as $\phi_{R/L}$. We then discretize the continuous field and obtain the overlap between the source and the pseudospin eigenstates as $SP_{\pm,R/L} = |\langle d_\pm | \phi_{R/L} \rangle|^2 /$ $\text{Max}(|\langle d_+ | \phi_{R/L} \rangle|^2, |\langle d_- | \phi_{R/L} \rangle|^2)$. Numerical calculations show that $SP_{+,R} = 1.63 \times 10^{-5}$, $SP_{-,L} = 1$ for RCP source and $SP_{+,R} = 1$, $SP_{-,L} = 1.63 \times 10^{-5}$ for LCP source, which reveal the pseudospin-dependent character of the source.

**Experiments.** In our experiments, the microwave near-field measurement system mainly consists of two parts: the vector network analyzer (Agilent E5063A) and the 3D near-field scanning platform. Three microwave antennas (with 0, $2\pi/3$, $4\pi/3$ degree phase delay) is mounted on the bottom of the aluminum plate through drilled holes. When the microwave probe antenna moves horizontally driven by a

stepper motor (with 2-mm step size), the $z$ component of the electric field of the TM modes in the frequency range of our interest (covering from 9.000 to 10.500 GHz) can be measured. To map the measured electric field distribution at selected frequencies more clearly, we define $(E_z/E_0)^2$ as variate to the plot, which is processed by MATLAB.

## Data availability

The data that support the plots within this work and other related findings are available from the corresponding authors upon reasonable request.

## Code availability

Numerical simulations in this work are all performed using the 2D radio frequency (RF) module of a commercial finite-element simulation software (COMSOL MULTIPHYSICS 5.4). All related codes can be built with the instructions in the Methods section.

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

## Acknowledgements

This work was financially supported by National Key R&D Program of China (Grant Nos. 2018YFA0306202 and 2017YFA0303702), National Natural Science Foundation of China (Grant Nos. 11625418, 51732006, 11674166, and 11774162) and China Post-doctoral Science Foundation Funded Project (Project No. 2019M661784).

## Author contributions

M.H.L., P.Z., and B.X. conceived the idea. B.X. did the theoretical analyses. G.S. and L.H. did the experiments. H.F.W. and G.S. performed the numerical simulations. B.X., H.F.W., F.L., and S.Y.Y. did the data analyses. Z.W. and Y.F.C. guided the research. All authors contributed to discussions of the project. B.X., H.F.W., G.S., P.Z., and M.H.L. prepared the paper with revisions from other authors.

## Competing interests

The authors declare no competing interests.
