## [Peer Review File · Nature Communications]

Reviewers' Comments:

Reviewer #1:

Remarks to the Author:

The authors took responsibility to answer the comments and described their position very well. I am thankful for the complete and visual answer that shows the serious approach of the authors. The authors revised their manuscript and added necessary information for readers. Also, I was very inspired by the proposed possible application. Unfortunately, the authors did not include any possible results confirming the real applications (and advantages) of the topological system in hand as a device in the revised version of the manuscript, thus focusing only on fundamental aspects.

In terms of fundamental aspects, and in general, in terms of novelty, I still have some reservations regarding this manuscript.

In the paper [J.Noh et al., Nat. Photon. 12, 408, 2018] higher-order topological corner states were observed experimentally for honeycomb. Furthermore, chiral eigenvalues were calculated for all types of corners in the system, showing different chirality and opposite signs of the topological index at the adjacent corners. Consequently, it seems that the novelty (in terms of authors' "first time" and "brand new concept") of the author's work concludes in the experimental demonstration of this chirality. That is an important achievement, but rather technical. The second comment is regarding the HO-QSHE. It seems that polarized coupling of topological corner modes goes through evanescent "tails" of topological edge states.

I see the work quite interesting from the experimental point of view, and I would recommend this paper to be published in a more specialized photonics journal.

Reviewer #2:

Remarks to the Author:

This paper reports the first observation of the pseudospin degree of freedom in higher order topological insulators. This effect is carefully characterized its potential applications are evident. Thus, this work expands the field of topological optics and will find application is topological insulators with different types of excitation. The impact of this work is well described in the conclusion. I therefore support the paper for publication in Nature Communications.

I still feel the work needs careful editing. I illustrate this with some comments on the concluding paragraph :

- In conclusion, we theoretically propose and experimentally realize a spinful higher-order photonic topological insulator in which both 1D edge states and 0D corner states are observed

◇

In conclusion, we theoretically propose and experimentally realize a spinful higher-order photonic topological insulator with both 1D edge states and 0D corner states.

- We achieve a directional localization of pseudospin polarized corner states of microwaves, denoted as the photonic higher-order version of the QSHE.

◇

We achieve the directional localization of pseudospin polarized corner states for microwave radiation, which are the photonic higher-order version of the QSHE.

- are influenced by the finite size effect and quality and the precision of the fabricated samples.

◇

are influenced by the finite size and the precision of the fabricated samples

- The difference in the frequencies between the simulation and experiment are induced by the finite height of the dielectric rods in the experiment.

◇

The differences in the frequencies between the simulation and experiment are due to the finite height of the dielectric rods in the experiment.

- Our work paves the way

◇

Our work opens the way

- The directional localization of 0D microwaves

◇

The directional localization of microwave radiation in 0D

- Moreover, since our implementation is simple and based on the dielectric material, it can be directly extended to optical frequencies by using silicon-based coupled resonator optical waveguides and femto-lasing direct writing methods.

◇

Since our implementation is simple and based on the dielectric material, this work can be extended to optical frequencies using silicon-based coupled resonator optical waveguides written with femto-second lasers.

- When quantum optics are considered, the single photons or multi-photons may have anomalous quantum behaviors which are related to the pseudospin degree of freedom such as the spinful quantum walk, spinful quantum correlations and entanglements

◇

It would then be possible to explore the quantum behavior of single or multiple photons which are related to the pseudospin degree of freedom, such as the spinful quantum walk, spinful quantum correlation and entanglement.

- If we couple our system to quantum emitters, the spinful corner states may support chiral emission of photons and potential applications in quantum simulations.

◇

When coupled to quantum emitters, the spinful corner states may support chiral emission of photons with potential applications in quantum simulations.

- Although our implementation is based on photonic crystals, we expect to observe a similar effect in other classical wave systems such as the phononic crystals, surface polariton plasmon, and electric circuit. {there is a lack of parallelism in the items listed}

◇

Although our implementation is based on photonic crystals, we expect to observe a similar effect in other periodic classical wave systems, such as the phononic crystals, and with excitations such as, surface polariton plasmon, and in electronic circuits.

Response Letter for “Higher-order Quantum Spin Hall Effect in a Photonic Crystal”

We appreciate the time and effort that editors and referees have put into the review process for our manuscript entitled "Higher-order Quantum Spin Hall Effect in a Photonic Crystal".

We are very grateful to all the referees for their valuable suggestions and important questions, which encourage us to clarify the novelty of our work and the difference between our work and previous works more clearly. We have addressed all the suggestions and comments in the Response Letter and provided a more detailed description of the application in the revised manuscript (marked in blue) and improved the English of our manuscript thoroughly (marked in blue). We believe that the revised manuscript is suitable for publication in Nature Communications.

Reviewer #1 (Remarks to the Author):

The authors took responsibility to answer the comments and described their position very well. I am thankful for the complete and visual answer that shows the serious approach of the authors. The authors revised their manuscript and added necessary information for readers. Also, I was very inspired by the proposed possible application. Unfortunately, the authors did not include any possible results confirming the real applications (and advantages) of the topological system in hand as a device in the revised version of the manuscript, thus focusing only on fundamental aspects.

Reply: We thank the reviewer for his acknowledgment of our previous response. Indeed, our work which firstly proposes and realizes the higher-order quantum spin Hall effect (HO-QSHE) in a photonic crystal is fundamentally novel and important. In terms of applications, besides those potential applications of our work when combined with other fields such as 2D luminescent materials, topological quantum optics, and topological lasers as discussed in the conclusion section, we can also see some direct applications from our results. As shown in Figs. 4e and 4f, we demonstrate a directional localization of topological polarized microwaves. This may enable robust and controllable microwave polarized probe and imaging. When considering the out-of-plane radiation, we may realize a topological microwave antenna that has the directional selection of emission of signals with a high-quality factor, low-Ohmic loss, and topological robustness. However, we think that the demonstration of these novel applications belongs to other stories and it is neither necessary nor proper to include them in the current manuscript. To further address the importance of our results, we have added more detailed discussions on the potential applications.

Revisions: We have added more detailed discussions on the potential applications as following,

“...As shown in Figs. 4e and 4f, we demonstrate a directional localization of topological polarized microwaves. This may enable robust and controllable microwave polarized probe and imaging. When considering the out-of-plane radiation, we may realize a topological microwave antenna that has the directional selection of emission of signals with a high-quality factor, low-Ohmic loss, and

topological robustness...

”

In terms of fundamental aspects, and in general, in terms of novelty, I still have some reservations regarding this manuscript.

In the paper [J.Noh et al., Nat. Photon. 12, 408, 2018] higher-order topological corner states were observed experimentally for honeycomb. Furthermore, chiral eigenvalues were calculated for all types of corners in the system, showing different chirality and opposite signs of the topological index at the adjacent corners. Consequently, it seems that the novelty (in terms of authors' "first time" and "brand new concept") of the author's work concludes in the experimental demonstration of this chirality. That is an important achievement, but rather technical.

Reply: We thank the reviewer for raising this important question which encourages us to explain the difference between our work and the paper [J. Noh et al., Nat. Photon. 12, 408, 2018] more clearly. We respectfully disagree with the reviewer that our work is a demonstration of the chirality of corner states in [J. Noh et al., Nat. Photon. 12, 408, 2018] and is a technical achievement. The reasons are provided below.

- (1) The observed higher-order quantum spin Hall effect (HO-QSHE) is **not** the manifestation of the chirality of corner states defined in [J. Noh et al., Nat. Photon. 12, 408, 2018]. The chirality of an eigenstate is defined as its common eigenvalue of the chiral operator which is also the sublattice symmetry operator defined in Eq. S25 in [J. Noh et al., Nat. Photon. 12, 408, 2018]. This chirality plays a crucial role in determining whether a state is pinned at the zero energy or not (see Section D of the Supplementary Information in [J. Noh et al., Nat. Photon. 12, 408, 2018]). However, the HO-QSHE, or more specifically the pseudospins we defined in our photonic crystal is related to the C_6 symmetry in a unit cell which has nothing to do with the chiral symmetry. In fact, due to higher-order couplings such as the next-nearest coupling in our photonic crystals, the chiral symmetry is naturally broken in our photonic crystals (as shown in Fig. S1 in our SI, the band structures are not symmetric with respect to the zero energy and thus have no chiral symmetry).
- (2) In terms of the experimental novelty, in the femtosecond laser direct writing waveguides implemented in [J. Noh et al., Nat. Photon. 12, 408, 2018]), the authors did not consider a multipoint-injection of photons and control the relative phases between the six waveguides in a unit cell. Thus they can't observe the pseudospin degree of freedom, needless to say, the unprecedented HO-QSHE. We expect the future realization of HO-QSHE in the femtosecond laser direct writing waveguides (although it would be very difficult.) which may have potential applications in topological quantum optics.

The second comment is regarding the HO-QSHE. It seems that polarized coupling of topological corner modes goes through evanescent "tails" of topological edge states.

Reply: Yes, the HO-QSHE is related to the polarized coupling of topological modes goes through evanescent "tails" of topological edge states. **However, to realize HO-QSHE, one must also introduce the pseudospin-momentum locking.** Without considering the pseudospin-momentum

locking, the LCP source may localize the waves at left and (or) right corners, not at the single left corner (the same situation for RCP source). Therefore, the phrase “polarized coupling of topological corner modes goes through evanescent "tails" of topological edge states” is inadequate to describe our observation and we think the terminology “HO-QSHE” perfectly characterized this phenomenon with directional localization due to the pseudospin-momentum locking.

I see the work quite interesting from the experimental point of view, and I would recommend this paper to be published in a more specialized photonics journal.

Reply: We thank the reviewer for his careful and insightful comments. We believe that with the above response and revisions, our manuscript is fundamentally novel and suitable for publication in Nature Communications.

Reviewer #2 (Remarks to the Author):

This paper reports the first observation of the pseudospin degree of freedom in higher order topological insulators. This effect is carefully characterized its potential applications are evident. Thus, this work expands the field of topological optics and will find application in topological insulators with different types of excitation. The impact of this work is well described in the conclusion. I therefore support the paper for publication in Nature Communications.

Reply: We thank the reviewer for supporting our work for publication in Nature Communications.

I still feel the work needs careful editing. I illustrate this with some comments on the concluding paragraph:

- In conclusion, we theoretically propose and experimentally realize a spinful higher-order photonic topological insulator in which both 1D edge states and 0D corner states are observed

In conclusion, we theoretically propose and experimentally realize a spinful higher-order photonic topological insulator with both 1D edge states and 0D corner states.

- We achieve a directional localization of pseudospin polarized corner states of microwaves, denoted as the photonic higher-order version of the QSHE.

We achieve the directional localization of pseudospin polarized corner states for microwave radiation, which are the photonic higher-order version of the QSHE.

- are influenced by the finite size effect and quality and the precision of the fabricated samples.

are influenced by the finite size and the precision of the fabricated samples

- The difference in the frequencies between the simulation and experiment are induced by the finite height of the dielectric rods in the experiment.

The differences in the frequencies between the simulation and experiment are due to the finite height of the dielectric rods in the experiment.

- Our work paves the way

Our work opens the way

- The directional localization of 0D microwaves

The directional localization of microwave radiation in 0D

- Moreover, since our implementation is simple and based on the dielectric material, it can be directly extended to optical frequencies by using silicon-based coupled resonator optical waveguides and femto-lasing direct writing methods.

Since our implementation is simple and based on the dielectric material, this work can be extended to optical frequencies using silicon-based coupled resonator optical waveguides written with femto-second lasers.

- When quantum optics are considered, the single photons or multi-photons may have anomalous quantum behaviors which are related to the pseudospin degree of freedom such as the spinful quantum walk, spinful quantum correlations and entanglements

It would then be possible to explore the quantum behavior of single or multiple photons which are related to the pseudospin degree of freedom, such as the spinful quantum walk, spinful quantum correlation and entanglement.

- If we couple our system to quantum emitters, the spinful corner states may support chiral emission of photons and potential applications in quantum simulations.

When coupled to quantum emitters, the spinful corner states may support chiral emission of photons with potential applications in quantum simulations.

- Although our implementation is based on photonic crystals, we expect to observe a similar effect in other classical wave systems such as the phononic crystals, surface polariton plasmon, and electric circuit. {there is a lack of parallelism in the items listed}

Although our implementation is based on photonic crystals, we expect to observe a similar effect in other periodic classical wave systems, such as the phononic crystals, and with excitations such as, surface polariton plasmon, and in electronic circuits.

Reply: We thank the reviewer very much for his careful editing of our manuscript. We have carefully improved the manuscript thoroughly (marked in blue) to make sure it meets the high standard for publication in Nature Communications.

Reviewers' Comments:

Reviewer #1:

Remarks to the Author:

The authors presented some clarifications related to the raised questions and added appropriate corrections to the text.

It should be noted here that still, the authors did not present any possible results confirming the real applications (and advantages) of the topological system in hand, which definitely would make the paper stronger.

Reviewer's Comments:

The authors presented some clarifications related to the raised questions and added appropriate corrections to the text.

It should be noted here that still, the authors did not present any possible results confirming the real applications (and advantages) of the topological system in hand, which definitely would make the paper stronger.

Reply: We thank the reviewer for his acknowledgment of our previous response. We agree with the reviewer that further experimental results confirming more applications (and advantages) of the topological systems will make the paper stronger. Indeed, we are currently working on achieving novel topological optical devices based on the higher-order quantum spin Hall effect. We sincerely believe that our manuscript which first proposed the higher-order quantum spin Hall effect in a photonic crystal is fundamentally important and is suitable for publication in Nature Communications.